# Photochromic and Luminescent Properties of a Salt of a Hybrid Molecule Based on C_60_ Fullerene and Spiropyran—A Promising Approach to the Creation of Anticancer Drugs

**DOI:** 10.3390/molecules28031107

**Published:** 2023-01-22

**Authors:** Artur A. Khuzin, Dim I. Galimov, Liliya L. Khuzina

**Affiliations:** Institute of Petrochemistry and Catalysis, Ufa Federal Research Center of the Russian Academy of Sciences, 141 Oktyabrya Prospect, 450075 Ufa, Russia

**Keywords:** fullerene, spiropyran, merocyanine, photochromism, spectral–kinetic properties, luminescent properties, photodegradation

## Abstract

For the first time a pyrrolidinofullerene salt containing a spiropyran group and an ammonium group, capable of reversibly reacting to UV radiation, has been synthesized. Photoinduced reactions of the synthesized compounds were studied using absorption and luminescence spectroscopies, spectral and kinetic characteristics were measured. The hybrid molecule was found to exhibit intrinsic fluorescence even in the spirocyclic form. The C_60_ derivative showed a higher stability and better spectral and luminescent properties than the precursor.

## 1. Introduction

Currently, photochromic compounds with controlled properties are of interest for the design of various materials with specified properties. Among the broad spectrum of organic photochromic systems, spiropyrans are the most interesting representatives, since their spectral, photochemical, and luminescence characteristics can be varied over fairly broad ranges. The possibility of interconversion between two isomers of spiropyrans under the action of external stimuli [1,2,3] is of considerable interest for the various applications in science and technology. Owing to their unique properties, spiropyrans are used for the manufacture of photochromic glasses [4] and as chemosensors [5], regulators of chemical reactions [6], optoelectronic and holographic devices [7], memory cells [8], regulators of membrane permeability [9], and molecular engines [10].

Among the relevant studies, of considerable importance are works aimed at the development of nanomaterials with controlled properties. Investigations of the covalent binding of carbon clusters to photochromic compounds resulted in the design of conceptually new materials for various fields of science and technology [11].

It is known from the literature that ionic liquids with long alkyl side chains can be incorporated into lipid bilayers and thus disrupt the integrity of eukaryotic cell membranes, and after the subsequent distribution in the cell they are able to trigger cell apoptosis by the mitochondrial pathway [12,13,14]. Merocyanines formed upon the photochemical isomerization are a type of ionic compound and their binding to fullerene C_60_ gives hope that the resulting hybrid molecules would possess antitumor properties. This assumption is supported by the fact that, owing to high lipophilicity of the fullerene cage, hybrid molecules rapidly penetrate cell membranes, thus accessing cell organelles, which opens the way to the design of targeted drug delivery systems [15,16,17]. Moreover, given the fact that in recent years the popularity of using spiropyrans as vectors for targeted drug delivery has increased, this idea looks quite realistic [18,19,20]. Besides, the luminescent properties of merocyanine forms of C_60_ derivatives would allow not only visualization of the route and accumulation of these compounds in particular organelles of the target cells, but also the study of the cytotoxicity mechanisms using various cell lines. This is indicated by the fact that luminescence was detected upon photoirradiation of the hybrid molecule based on fullerene C_60_ and spiropyran [21]. Together with water solubility this would open up prospects for the design of effective photoswitchable antitumor agents based on these hybrids for the treatment of cancer in humans. Previously we reported on the synthesis of spiropyran salts and the study of their spectral and luminescent properties, which are of interest as antitumor compounds [22]. By the beginning of our work, the feasibility of this approach was evidenced by only one study addressing the photoswitchable cytotoxicity against Hek293 cancer cells for one spiropyran example [23].

The purpose of this work is to develop optimal conditions for the interaction of C_60_-fullerene with spirophotochrome, to obtain the corresponding hybrid adduct, as well as to study the spectral and luminescent properties in order to study the reversible isomerization of the synthesized compounds with the formation of ionic compounds (merocyanine forms). The presence of luminescent properties in the synthesized compounds will allow further study of the mechanism of their biological action on the studied cancer cells using fluorescence microscopy.

## 2. Results and Discussion

### 2.1. Chemistry

In view of the foregoing to attain our goal, that is, to prepare a new water-soluble hybrid compound, we chose spiropyran salt **1** containing several functional groups in the molecule as the model spirophotochrome. The desired spiropyran was synthesized using methods reported in the literature [22,23] according to Figure 1:

The subsequent synthesis of preparative amounts of the hybrid compound was performed using the Prato reaction, which includes 1,3–dipolar cycloaddition to C_60_ to azomethine ylides generated in situ by the reaction of aldehyde with sarcosine [24]. Thus, the cycloaddition of compound **1** to fullerene C_60_ in the presence of N–methylglycine at a temperature of 110 °C afforded fulleropyrrolidine **2** in 70% yield (Figure 2). Compound **2** was isolated from the reaction mixture by column chromatography (SiO_2_). Elution with toluene washed out unreacted toluene, and the subsequent elution with pyridine gave target adduct **2** with a purity of 99%.

The structure of compound **2** was established by 1D (^1^H and ^13^C) and 2D (H–HCOSY, HSQC, HMBC) NMR spectroscopy and high–resolution mass spectrometry (HRMS, see Appendix A).

In the ^1^H NMR spectrum of compound **2** in CDCl_3_ a singlet at 4.94 ppm and two doublets at 5.01 and 4.30 ppm with spin-spin coupling constants of 9.31 Hz correspond to protons at the pyrrolidine carbon atoms. Two singlets at 2.14 ppm and 2.92 ppm belong to protons of the methyl groups at the pyrrolidine and ammonium nitrogen atoms, respectively. The singlets with chemical shifts of 2.14 and 2.18 ppm are due to the protons of the two methyl groups of the indole ring. The characteristic doublet at 5.80 ppm with a spin-spin coupling constant of 10.36 Hz, corresponding to the CH group at the spiro atom in the pyran moiety, attests to the presence of a spirocyclic structure (see Appendix A).

In the ^13^C NMR spectrum of **2** in CDCl_3_, the pyrrolidine moiety gives rise to carbon signals at 70.05 (CH_2_), 83.66 (CH), and 40.16 ppm (N–CH_3_), which are correlated, in the HSQC experiment, with the proton signals at 5.01 (d, 1H, CH_2_), 4.30 (d, 1H, CH_2_), 4,94 (s; 1H, CH), and 2.91 (N–CH_3_) ppm (see Appendix A).

### 2.2. UV–Vis and Luminescent Studies

The next stage of our study addressed the photochromic and luminescent properties of compounds **1** and **2** in spiropyran (**1** and **2**) and merocyanine (**1′** and **2′**) forms in an aprotic solvent–tetrahydrofuran (THF), where some regularities were identified. THF was chosen as the solvent because of its moderate polarity, high dissolving capacity and inertness to the ammonium salts of **1** and **2**. The results of photophysical studies are shown in Table 1.

Study of the kinetic and spectral characteristics showed that on UV irradiation a solution of fulleropyrrolidine **2** in THF acquires an intense blue color (Appendix A), indicating conversion of the spirocyclic form to the merocyanine form. Similar changes are also characteristic of **1** (Appendix A). The absorption spectrum of compound **2′** (Figure 1) exhibits new strong absorption bands at 381 and 594 nm, apart from the bands at 255, 328, and 431 nm, characteristic of C_60_ cycloadducts. The close positions of the absorption bands in the spectra of **1′** (374, 602 nm) and **2′** (381, 594 nm) in the open–ring form implies that these bands are due to electron transitions only in the merocyanine moiety of the hybrid molecule, i.e., the photoinduced structural transformations do not affect the fullerene cage (Figure 1, inset). The absorption spectra of compounds **1** and **2** also contain one isosbestic point at 347 and 351 nm, respectively. The presence of an isosbestic point in the spectrum indicates a high rate of photoinduced opening of the pyran ring and the presence of only two types of compounds in the system: open and closed forms (i.e., the absence of intermediate forms).

Simultaneously while studying the kinetic characteristics of fulleropyrrolidine **2′** we found that its photochromic properties are considerably influenced by the π-conjugated system of the fullerene cage. First, this influence is manifested as a decrease in the rate of spontaneous dark bleaching (k_1_ = 0.014 s^−1^) and photobleaching (k_2_ = 0.018 s^−1^) of the merocyanine form of **2′** (Figure 2, Table 1) in comparison with that of **1** (0.028 and 0.031 s^−1^, respectively). Second, the light sensitivity of **2** (S = 0.87) is lower than that of precursor **1** (S = 1.30).

These differences are attributable to the re-absorption of the activating UV radiation by the fullerene cage, which intensely absorbs in the absorption region of the photochromic spiropyran moiety. This is indicated by the considerable difference between the photodegradation efficiency of compounds **2′** and **1′** (Table 1, Figure 3). It was found that hybrid compound **2′** is much more stable to irreversible phototransformations (τ_1/2_ > 700 s) than compound **1′** (τ_1/2_ = 64 s), which is consistent with published data [3,25].

Particular attention is warranted for analysis of the kinetic curves for the photodegradation of **2′** under irradiation with filtered UV light. It can be seen in Figure 3 that under UV irradiation of a THF solution of fulleropyrrolidine **2′** after the photoequilibrium has been established the photoinduced absorbance first monotonically decreases and then increases, thus giving rise to two additional maxima in the kinetic curve of photodegradation. Repeated experiments demonstrated that this shape of the kinetic curve is observed in all cases. This unexpected result can be attributed to the fact that long-term UV irradiation of a solution of compound **2′** induces irreversible photochemical transformations giving intermediate photoproducts characterized by a stronger absorption in the 450–650 nm range compared to fulleropyrrolidine **2′**. Subsequently we intend to perform a thorough analysis in order to prove the formation of particular photodecomposition products.

Simultaneously, some regularities of generation and deactivation of the electronically excited states upon photoirradiation were studied in relation to compounds **1** and **2**, and comparative analysis of the results was carried out. It was found that spiropyran **1** in the closed spirocyclic form does not exhibit photoluminescence (PL). However, fairly intense PL appears in the red region of the visible spectrum with a peak at 645 nm upon UV irradiation of a THF solution of compound **1′** at room temperature (Appendix A). The positions of peaks in the absorption spectra (Appendix A, curve a) and PL excitation spectra (Appendix A, curve b) of spiropyran **1′** virtually coincide, indicating that the observed PL is caused by radiative transitions in the merocyanine form.

The subsequent studies demonstrated that, unlike precursor **1**, fulleropyrrolidine **2** in THF possesses intrinsic fluorescence in the near–IR range with a diffuse maximum at 730–740 nm without activation by UV light (Figure 4).

The peaks in the PL spectra (λ_max_ = 734 nm) and PL excitation spectra (λ_max_ = 480 nm) of fulleropyrrolidine **2** are very similar in intensity and position to analogous peaks for buckminsterfullerene C_60_ (740 and 488 nm, respectively). This fact indicates that the luminescence of **2** in the closed form is due to the radiative S_1_→S_0_ transition between the fullerene electronic states (Figure 5).

It was shown that further UV irradiation of a THF solution of **2** leads to photoinduced isomerization of the spirocyclic form to the merocyanine form, accompanied by a pronounced change in the luminescent properties. The PL spectrum of the open–ring form of **2′** shows peaks at 632, 705, and 790 nm (Figure 6).

The close positions of the peaks in the PL spectra of **1′** and **2′** suggest that the short-wavelength component (632 nm) of the PL spectrum of merocyanine **2′** is due to fluorescence of the merocyanine moiety of the hybrid molecule, while the long–wavelength band (705, 790 nm) corresponds to fluorescence of the fullerene cage (Figure 5). To confirm this assumption, we measured the excited state lifetimes (τ) of the spirocyclic and merocyanine forms of **1** and **2** at the PL peak wavelengths. The obtained data (Table 1) indicate that the luminescence of fulleropyrrolidine **2** is indeed due to the radiative transitions between the electronic states of the merocyanine and fullerene moieties. The measured lifetime of ^1^(**2′**) * at 640 nm (τ = 3.1 ns) is somewhat greater than that for ^1^(**1′**)* (τ = 2.7 nm), which attests to the brighter PL of the hybrid molecule. This is also confirmed by estimation of the relative PL intensities by calculating the areas under the PL curves. It was found that the PL intensity of **2′** is almost 7 times higher for the open form than for the closed form and 4 times higher than the PL of precursor **1′** (Table 1).

## 3. Materials and Methods

The detailed procedure of the synthesis and characterization of the products are given in the Appendix A.

## 4. Conclusions

Thus, we performed the first synthesis of fulleropyrrolidine salt containing a spiropyran moiety and an ammonium group and studied the photochromic and luminescent properties of the product. The results lead to the conclusion that the hybrid compound, like its precursor, shows positive photochromism. It was found that unlike the starting spiropyran **1**, fulleropyrrolidine derivative **2** in the spirocyclic form exhibits intrinsic fluorescence. The derivative of C_60_ was found to be much more stable to irreversible photochemical transformations than the precursor and to possess enhanced spectral and luminescent properties. Based on the results obtained in this work and known literature data on the high solubilizing ability of fullerene-containing compounds [26,27,28,29], in the future it is planned to study the selective cytotoxicity of photochromic fulleropyrrolidines, their effect on the cell cycle and the ability to induce apoptosis using modern methods of flow cytometry on various tumor cells.

## Data Availability

Not applicable.

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
