# Peer review of "Photochromic and Luminescent Properties of a Salt of a Hybrid Molecule Based on C60 Fullerene and Spiropyran—A Promising Approach to the Creation of Anticancer Drugs"

_molecules, 2023, doi:10.3390/molecules28031107_

Round 1

Reviewer 1 Report

This submission reports the design and preparation of a photochromic and luminescent hybrid molecule based on C60 fullerene and spiropyran. After the identification of the synthesized hybrid molecule, the optical properties of the synthesized spiropyran and hybrid molecule are investigated and compared. This manuscript could be published in Molecules journal after considering the following major revisions.

Q1. In the abstract, the significance of the present research is not presented clearly; please declare the consequence of current results over previous reports. The abstract should briefly describe the challenges in this field, provide hints of obtaining results and co-relate with your significant work.

Q2. Do not use lumped references such as [1-11], [12-20], and [21-51] in the introduction section. In addition, it is not necessary to use 30 references for convincing a paragraph. It is suggested to refer to some recent related review articles or books.

Q3. When referring to the features of spiropyran, and their applications in biomedical fields and drug delivery, some recent and relevant review articles are suggested to be cited properly instead of old references: Doi: 10.3389/fchem.2021.720087, 10.1021/acsomega.2c04969, 10.1016/j.jphotochemrev.2022.100487

Q4. Page 2 Line 65, it has been stated that fulleropyrrolidine 2 was obtained with 55% yield, while in Scheme 1. Synthesis of compound 2, it has been stated 70%:?

Q5. The authors should consider a unique name for spiropyran (for example SP and MC) and hybrid molecules (SP@Fl and MC@Fl) in the whole manuscript instead of multiple names such as, fulleropyrrolidine 2, hybrid 2, compound 2, and 2 which are too confusing. Please revise the manuscript thoroughly.

Q6. For Figures 2-6, It is suggested to utilize symbols on curves and legend options to show different absorption and PL excitation and PL spectra instead of 1, 2, and 3.

Q7. Why did the authors choose tetrahydrofuran as the solvent??, please describe in the manuscript. Does the optical behavior of compound 1 and compound 2 change by replacing the solvent??

8 Please add the fluorescent and naked-eye visual pictures of compound 1 and compound 2, before and after UV irradiation to show the photoisomerization process and clarify their differences.

Q9. The title is “Photochromic and luminescent properties of a salt of a hybrid molecule based on C60 fullerene and spiropyran – a Promising Approach to the Creation of Anticancer Drugs”, however, I could not find any analyses on or description about the advantages of the synthesized hybrid molecule as an anticancer drug in the results and discussion??

Reviewer 2 Report

molecules-2143739

Review for article: 

« Photochromic and luminescent properties of a salt of a hybrid molecule based on C60 fullerene and spiropyran – a Promising Approach to the Creation of Anticancer Drugs»

by Artur Al'bertovich Khuzin, Dim Irshatovich Galimov, Liliya Linatovna Khuzina

In Molecules (ISSN 1420-3049).

Round 1

This work demonstrated the possibilities of pyrrolidinofullerene salt containing a spiropyran moiety and an ammonium group, which was synthesized for the first time. These molecules exhibit intrinsic fluorescence even in the spirocyclic form. The С60 derivative showed a higher stability and better spectral and luminescent properties than the precursor.

The article was presented in a weakly structured manner, with poor organization. Unfortunately, several statements within have weak evidence. Therefore, the referee suggested that the manuscript should be considerably improved in a major revision. The following is a list of specific concerns. 

-       Introduction. 

The introduction aims to attract the reader to the Molecules (ISSN 1420-3049) by providing contrast to other similar publications.

There are no good, cited works since 2020 (only one for 2020, one for 2021, and one for 2022). A more current reference list should be expanded and added. There are no works cited since late 2022 and 2023. Please provide a more representative reference list.

The objectives of the study should be stated in a separate paragraph.

Use separate bibliography references in all texts without combination to much more clarity and understandings of this text. 

It will be better to present Jablonski diagram for these compounds.

-       RESULTS AND DISCUSSION

There is no data about the solubility of these substances.

As well, there is no stability data even by UV/vis time changes in spectra.

Figure 1. Please assign any peaks. Also, isosbestic point at 340 nm should be discussed.

-       Conclusions

The article’s conclusion should be a detailed conclusion about the state of the field of science to date, with a brief description of achievements, shortcomings, and prospects for future development. It should be expanded. Works concerning the solubilization of fullerenes in water (aqueous media) should be added to the reference list. Some of them are presented below.

·      https://www.tandfonline.com/doi/abs/10.1080/1536383X.2014.998758;

·      https://www.sciencedirect.com/science/article/pii/S0167732218355685?via%3Dihub;

·      https://www.sciencedirect.com/science/article/pii/S1350417721000754?via%3Dihub;

·      https://pubs.acs.org/doi/full/10.1021/cr3005026.

-       Materials and Methods

Line 202: The detailed procedure of the synthesis and characterization of the products are given in supplementary materials.

I couldn’t find it in the SuSy or elsewhere.

Style guide issues

Line 118: It should be minus (–), not dash (-) etc.

English spelling should be double-checked. 

Round 2

Reviewer 1 Report

This manuscript was revised according to the comments, and the quality is much improved. I recommend its publication.

Reviewer 2 Report

molecules-2143739

Review for article:

«Photochromic and luminescent properties of a salt of a hybrid molecule based on C60 fullerene and spiropyran – a Promising Approach to the Creation of Anticancer Drugs»

by Artur Al'bertovich Khuzin, Dim Irshatovich Galimov, Liliya Linatovna Khuzina

In Molecules (ISSN 1420-3049).

Round 2

The authors addressed well my comments. I would recommend its acceptance after correction following statement.

But, it should be change in section

1.     Line 114, 118.

The use of the term optical density is discouraged. It should be used as ∆Aphot/Amax

(or as an Abs). (PAC, 1996, 68, 2223. (Glossary of terms used in photochemistry (IUPAC Recommendations 1996)) on page 2257)